# Estimation of Longitudinal Force, Sideslip Angle and Yaw Rate for Four-Wheel Independent Actuated Autonomous Vehicles Based on PWA Tire Model

**DOI:** 10.3390/s22093403

**Published:** 2022-04-29

**Authors:** Xiaoqiang Sun, Yulin Wang, Weiwei Hu

**Affiliations:** Automotive Engineering Research Institute, Jiangsu University, Zhenjiang 212013, China; 2222004051@stmail.ujs.edu.cn (Y.W.); 2221904073@stmail.ujs.edu.cn (W.H.)

**Keywords:** autonomous vehicles, four-wheel independently actuated (FWIA), square-root cubature kalman filter (SCKF), vehicle lateral dynamics, piecewise affine (PWA) identification

## Abstract

This article introduces an efficient and high-precision estimation framework for four-wheel independently actuated (FWIA) autonomous vehicles based on a novel tire model and adaptive square-root cubature Kalman filter (SCKF) estimation strategy. Firstly, a reliable and concise tire model that considers the tire’s nonlinear mechanics characteristics under combined conditions through the piecewise affine (PWA) identification method is established to improve the accuracy of the lateral dynamics model of FWIA autonomous vehicles. On this basis, the longitudinal relaxation length of each tire is integrated into the lateral dynamics modeling of FWIA autonomous vehicle. A novel nonlinear state function, including the PWA tire model, is proposed in this paper. To reduce the impact of the uncertainty of noise statistics on the estimation accuracy, an adaptive SCKF estimation algorithm based on the maximum a posteriori (MAP) criterion is proposed in the estimation framework. Finally, the estimation accuracy and stability of the adaptive SCKF algorithm are verified by the co-simulation of CarSim and Simulink. The simulation results show that when the statistical characteristics of noise are unknown and the target state changes suddenly under critical maneuvers, the estimation framework proposed in this paper still maintains high accuracy and stability.

## 1. Introduction

With the sharp increase in vehicle ownership, the impact of pollution on the human environment is becoming more and more serious [1,2]. Electric vehicles (EVs) have significant advantages in energy-saving and environmental protection, which has been a wide concern of society [3]. In particular, the four-wheel independent actuated (FWIA) electric vehicle has been said to be an effective scheme, having no complex transmission structure and four-wheel motors that can be controlled independently. It can not only actively adjust the torque of each wheel but also has high torque flexibility and high control precision. As an excellent platform for autonomous technology, FWIA autonomous vehicles have aroused widespread interest in researchers [4,5].

In recent years, with the application of advanced control systems such as adaptive cruise control (ACC) [6], direct yaw moment control (DYC) [7], lane-keeping assistance (LKS) [8], and lane departure warning (LDW) [9] to autonomous vehicles, dynamic control performance and driving safety has been greatly improved. In the closed-loop control architecture, the reliable estimation inputs of autonomous vehicle motion states have become the crucial premise of high-precision dynamic control of FWIA autonomous vehicles [10,11]. These parameters for the control system, i.e., yaw rate, sideslip angle, and longitudinal force, are the basis for path tracking and the lateral stability of autonomous vehicles. However, some state parameters are difficult to measure by vehicle-mounted sensors and must be measured by expensive sensors. In this condition, model-based vehicle state observers have attracted widespread attention and become the first choice for researchers [12]. High-precision vehicle models and advanced algorithms are essential for vehicle state estimation in practice.

As the only component of a vehicle that contacts the road and the vehicle, tires are a significant element in vehicle dynamics control [13]. The estimation of longitudinal force and lateral force plays a crucial role in the design of trajectory tracking and the lateral stability control system of autonomous vehicles. Under extreme conditions, the mechanical characteristics of tires show high nonlinearity. At this time, the accuracy of the tire model will have a great impact on the accuracy of parameter estimation [14,15]. However, complex expressions will bring great challenges to the computational efficiency of ECU [16]. In practical application, the expression of a tire model should be concise enough to ensure strong real-time performance. Based on the above analysis, it is important to build a high-precision and concise tire model. The relationship between tire longitudinal force and lateral force under combined conditions are highly coupled. It is difficult to achieve accurate results using the mechanism modeling method [17]. With the rapid development of computer technology, data-driven multi-model identification methods can be used as an effective way to break through the research bottleneck [18,19]. In this paper, a data-driven multi-input and multi-output piecewise affine identification method is used to approximate the nonlinear system through multiple affine sub-models, and the expression of the piecewise affine identification model is concise enough.

In recent years, more and more advanced theories have been applied in vehicle engineering. Many achievements have been reached in the estimation of vehicle yaw rate, sideslip angle, and four-wheel longitudinal force. A novel observer considering the uncertainty of the tire model under combined conditions is proposed in [20]. On this basis, the accuracy and robustness of the estimation algorithm were verified using a simulation of a 14-DoF Simulink vehicle model. In [21], a novel model of an electric wheel is proposed. Furthermore, the state function of longitudinal force is established by using the sliding mode observer, and an adaptive square-root cubature Kalman filter (SCKF) was designed to estimate the noise to improve the stability of the vehicle. Finally, the effectiveness of the proposed estimation algorithm was verified by CarSim-Simulink co-simulation. The advanced estimation strategies utilized for observer design in prior works include the extended Kalman filter (EKF) [22], unscented Kalman filter (UKF) [23], cubature Kalman filter (CKF) [24,25], and information fusion estimation method [26]. In [27], an uncertain singular vehicle model was established, which considers the time-varying characteristics of tire cornering stiffness and includes the singular vehicle dynamics model and the uncertainty of the model. On this basis, a robust sideslip angle observer has been established. In [28], Wang et al. proposed a robust unscented Kalman filter estimation algorithm to improve the robustness of outliers measured by sensors. The influence of noise was handled by a moving polynomial Kalman smoother. In [29], an integral correction fusion estimator based on the adaptive square-root CKF was proposed. The zero-point-reset method was used to correct the colored noise of sensors. The error caused by vehicle nonlinear dynamics can be compensated by the novel estimation strategy. In [30], a random walk square-root CKF estimation strategy considering vehicle parameter uncertainty was proposed. The effectiveness of the estimation of longitudinal tire force and lateral force was verified by CarSim-Simulink co-simulation. When the sudden change of target motion state leads to a mismatch between the vehicle model and the real-world or statistical characteristics of noise are unknown, the accuracy and stability of the traditional square-root cubature Kalman filter will be reduced [31,32]. Thus, an adaptive SCKF algorithm is proposed, which estimates the parameters using the maximum a posterior (MAP) criterion to improve the estimation stability of the SCKF algorithm.

This article introduces a novel estimation architecture for longitudinal force, yaw rate, and the sideslip angle of autonomous vehicles via an adaptive SCKF algorithm. Firstly, a novel tire model under combined conditions is obtained using the piecewise affine identification method, which has the advantages of high precision and conciseness. Secondly, a novel nonlinear state function, which considers the tire dynamics characteristics and the PWA tire model, is derived. Finally, an ASCKF algorithm is proposed to improve the dynamic adjustment and self-adaptation performance when dealing with uncertainty interference. Its effectiveness has been verified by co-simulation.

The rest of this paper is organized as follows. In Section 2, a 3-DoF vehicle model is introduced. Section 3 discusses how the novel PWA tire model was obtained based on piecewise affine identification is introduced. In Section 4, a novel nonlinear state-space function and an adaptive SCKF algorithm are proposed to estimate four-wheel longitudinal force, yaw rate, sideslip angle, longitudinal velocity, and lateral velocity. In Section 5, the effectiveness and practicability of the estimation architecture are verified by CarSim-Simulink co-simulation.

## 2. Vehicle Dynamics Modeling

To reduce the complexity of the vehicle dynamics model as much as possible, a 3-DoF vehicle model, which considers the longitudinal motion, lateral motion, and yaw motion, is established in Figure 1. The dynamics model can be expressed as follow [33]:
(1)$$\left\{\begin{array}{l} \begin{array}{c} {\dot{v}}_{x}=a_{x}+v_{y}r\\ {\dot{v}}_{y}=a_{y}-v_{x}r \end{array}\\ I_{z}\cdot \dot{r}=\left[\left(F_{xfl}+F_{xfr}\right)\sin\delta +\left(F_{yfl}+F_{yfr}\right)\cos\delta \right]l_{f}-\left(F_{yrl}+F_{yrr}\right)l_{r}\\ +\left[\left(F_{xfr}-F_{xfl}\right)\cos\delta +\left(F_{yfl}-F_{yfr}\right)\sin\delta \right]\frac{B}{2}+\left(F_{xrr}-F_{xrl}\right)\frac{B}{2} \end{array}\right.$$
where *δ* is the front wheel angle. *B* is the vehicle track width. *I_z_* is yaw inertia of the vehicle. *l_f_* and *l_r_* are the distances from the vehicle center to the front and rear axle. *v_x_*, *v_y_* and *r* are the longitudinal velocity, lateral, and yaw rate. *F_xij_* and *F_yij_* (*i* = *f* or *r*, *j* = *r* or *l*) represent the longitudinal forces and lateral forces of each tire. *a_x_* and *a_y_* denote the longitudinal acceleration and the lateral acceleration, which can be expressed as:(2)$$\left\{\begin{array}{l} a_{x}=\frac{1}{m}\left[\left(F_{xfl}+F_{xfr}\right)\cos\delta -\left(F_{yfl}+F_{yfr}\right)\sin\delta +F_{xrl}+F_{xrr}\right]\\ a_{y}=\frac{1}{m}\left[\left(F_{xfl}+F_{xfr}\right)\sin\delta +\left(F_{yfl}+F_{yfr}\right)\cos\delta +F_{yrl}+F_{yrr}\right] \end{array}\right.$$
where *m* is the vehicle mass. In this work, the vehicle sideslip angle *β* and longitudinal slip *λ* of each tire can be given as follows [34]:(3)$$\beta =\arctan(\frac{v_{y}}{v_{x}})\approx \frac{v_{y}}{v_{x}}$$
(4)$$\left\{\begin{array}{l} \lambda_{fl}=\frac{\omega_{fl}R-v_{fl}}{v_{fl}},\;\lambda_{fr}=\frac{\omega_{fr}R-v_{fr}}{v_{fr}}\\ \lambda_{rl}=\frac{\omega_{rl}R-v_{rl}}{v_{rl}},\;\lambda_{rr}=\frac{\omega_{rr}R-v_{rr}}{v_{rr}} \end{array}\right.$$
where *w_ij_*, *λ_ij_*, and *v_ij_* (where “*fl*”, “*fr*”, “*rl*”, “*rr*” stand for the left front wheel, the right front wheel, the left rear wheel, and the right rear wheel) are the angular velocity of each wheel, the tire longitudinal slip, and four wheels center velocity. *R* is the wheel radius. 

Based on wheel dynamics analysis, the velocity of each wheel center and tire sideslip angles can be expressed as follows [35]:(5)$$\left\{\begin{array}{l} v_{fl}=\left(v_{x}-\frac{B}{2}r\right)\cos\delta +\left(v_{y}+l_{f}r\right)\sin\delta \\ v_{fr}=\left(v_{x}+\frac{B}{2}r\right)\cos\delta +\left(v_{y}+l_{f}r\right)\sin\delta \\ v_{rl}=v_{x}-\frac{B}{2}r;\;v_{rr}=v_{x}+\frac{B}{2}r \end{array}\right.$$
(6)$$\left\{\begin{array}{l} \alpha_{fl}=\delta -\arctan\left(\frac{v_{y}+l_{f}r}{v_{x}-Br/2}\right);\alpha_{fr}=\delta -\arctan\left(\frac{v_{y}+l_{f}r}{v_{x}+Br/2}\right)\\ \alpha_{rl}=-\arctan\left(\frac{v_{y}-l_{r}r}{v_{x}-Br/2}\right);\alpha_{rr}=-\arctan\left(\frac{v_{y}-l_{r}r}{v_{x}+Br/2}\right) \end{array}\right.$$
where *α_fl_*, *α_fr_*, *α_rl_*, and *α_rr_* are the tire sideslip angles.

Unlike the traditional linear tire model, the nonlinear relationship between tire force and their influencing factors, i.e., the tire sideslip, angle *α*, and the tire longitudinal slip *λ*, is fully considered in this work. A novel longitudinal and lateral tire forces model which obtained by piecewise affine identification can be expressed as follows:(7)$$\left\{\begin{array}{ll} F_{xfl}=f_{PWA}(\alpha_{fl},\lambda_{fl},\mu ), & F_{xfr}=f_{PWA}(\alpha_{fr},\lambda_{rl},\mu )\\ F_{xrl}=f_{PWA}(\alpha_{rl},\lambda_{rl},\mu ), & F_{xrr}=f_{PWA}(\alpha_{rr},\lambda_{rr},\mu )\;\\ F_{yfl}=g_{PWA}(\alpha_{fl},\lambda_{fl},\mu ), & F_{yfr}=g_{PWA}(\alpha_{fr},\lambda_{rl},\mu )\\ F_{yrl}=g_{PWA}(\alpha_{rl},\lambda_{rl},\mu ), & F_{yrr}=g_{PWA}(\alpha_{rr},\lambda_{rr},\mu ) \end{array}\right.$$
where *μ* is road adhesion coefficient.

## 3. PWA Modeling of Tire Mechanical Properties under Combined Conditions

The longitudinal and lateral forces of the tire have a complex coupling relationship under extreme conditions, which brings a big challenge for estimating the tire force of the FWIA autonomous vehicle. The estimation of tire force in high precision a crucial to improving the control performance of path tracking and lateral stability. Thus, this work has great significance.

In this section, the experimental data of tire force under combined conditions can be obtained through the bench tests in Figure 2. During the test procedure, the rolling plate with material properties similar to a road surface was selected for a bench test and the tire was driven at a constant speed on the rolling plate. The Kong Hui automobile technology (KHAT) low-speed flat tire mechanical characteristic bench test was adopted in this experiment. The adopted tire was the DOUBLECOIN RT500 215/75 R17.5 127/124M. The specific parameter settings of the tire bench tests are shown in Table 1. 

The test results of the tire’s nonlinear mechanical properties under combined conditions are shown in Figure 3. As can be seen from these two figures, the highly nonlinear relationships between the tire forces and their influence factors (tire sideslip angle *α*, longitudinal slip *λ*) are manifested in the irregular surfaces’ form. This illustrates the complex coupling relationship between longitudinal force and the lateral force of the tire under combined conditions. In this work, the PWA identification is regarded as an effective way to achieve the modeling of the tire’s nonlinear mechanical properties. The irregular surfaces are approximated by several affine submodels, which are manifested in the form of flat surfaces. Based on the obtained experimental data, which reflects the tire’s nonlinear mechanical properties accurately, the several affine models and their switching rules can be constructed by PWA identification.

This work, which is regarded as an identification problem of the three-dimensional PWA system, can be divided into three steps: (1) the data clustering; (2) the parameter estimation of the affine submodels; (3) the calculation of the hyperplane coefficient matrices.

The model of a nonlinear dynamical system in the PWA form can be expressed as [36,37]:(8)$$y_{j}=\left\{\begin{array}{cc} \theta_{1}{}^{T}\left[\begin{array}{c} x_{j}\\ 1 \end{array}\right], & \mathrm{if}\;\;\;x_{j}\in \chi_{1}\\ \vdots & \vdots \\ \theta_{s}{}^{T}\left[\begin{array}{c} x_{j}\\ 1 \end{array}\right], & \mathrm{if}\;\;\;x_{j}\in \chi_{s} \end{array}\right.$$
where *y_j_* ∈ **R***^p^* denotes the PWA system output, *θ_i_* (*i* = 1, …, *s*) are the parameters of each affine submodel, and *x_j_* ∈ **R***^n^* represents the system regression vector. It consists of the system past inputs and outputs:(9)$$x_{j}={\left[y_{j-1},y_{j-2}^{},\cdots ,y_{j-n_{y}}^{},u_{j-1},u_{j-2}^{},\cdots ,u_{j-n_{u}}^{}\right]}^{T}$$
where *n_y_* and *n_u_* are the orders of PWA model, *u_j_* ∈ **R***^m^* denotes the system inputs, and *n* = *pn_y_* + *mn_u_*. *χ_i_* (*i =* 1, *…*, *s*) represent the complete partitions of the regressor set *χ*, and each region *χ_i_* is a convex polyhedral subset represented in the following form: (10)$$\chi_{i}=\{F_{i}x_{j}+g_{i}\leq 0\}$$
where *F_i_* and *g_i_* are the hyperplane coefficient matrices.

### 3.1. Data Clustering

The original data is divided into *s* disjoint clusters through data clustering. The Gaussian mixture model has excellent mathematical properties and computational performance. It is adopted in this paper [38]. The *N* data samples can be assumed as:(11)$$z_{{}_{j}}=\left[\begin{array}{l} x_{j}\\ y_{j} \end{array}\right]\in R^{n+p},\;j=1,2,\cdots ,N$$

The probability density of data sample *z_j_* can be given by: $$p(z;\Phi )=\sum_{i=1}^{s}\alpha_{i}p_{i}(z;\mu_{i},\Sigma_{i})$$
(12)$$\left\{\begin{array}{l} \Phi \;:=(\alpha ,\mu ,\Sigma )\\ \alpha \;:=(\alpha_{1},\alpha_{2},\cdots ,\alpha_{s}),\;\sum_{i=1}^{s}\alpha_{i}=1\\ \mu \;:=(\mu_{1},\mu_{2},\cdots ,\mu_{s})\\ \Sigma \;:=(\Sigma_{1},\Sigma_{2},\cdots ,\Sigma_{s}) \end{array}\right.$$
where *μ* represents *n* + *p*-dimensional mean vector. Σ is (*n* + *p*) × (*n* + *p*)-dimensional covariance matrices. Multivariate gaussian density *p_i_* is defined as:(13)$$p_{i}(z;\mu_{i},\Sigma_{i})=\frac{1}{{\left(2\pi \right)}^{(n+p)/2}{\left[\det(\Sigma_{i})\right]}^{1/2}}\times \exp\left\{-\frac{1}{2}{\left(z-\mu_{i}\right)}^{T}\Sigma_{i}^{-1}(z-\mu_{i})\right\},i=1,2,\cdots ,s$$
where the optimal parameter *Φ* can be obtained as follows
(14)$$P(j\in \Gamma_{i})=\frac{\alpha_{i}p_{i}(z_{j};\mu_{i},\Sigma_{i})}{p(z_{j};\Phi )}$$

The maximum-likelihood (ML) estimation is adopted in this work to find the optimal *Φ*. It can be expressed as follows:(15)$$L(\Phi )=\sum_{j=1}^{N}\ln p(z_{j};\Phi )=\sum_{j=1}^{N}\ln\left(\sum_{i=1}^{s}\alpha_{i}p_{i}(z_{j};\mu_{i},\Sigma_{i})\right)$$

Furthermore, the execution process of the EM algorithm which contains the expectation step (E-step) and the maximization step (M-step) is shown in Algorithm 1.
**Algorithm 1**. The Execution Process of the EM AlgorithmStep 1: Initialize Φ(0)=(α(0),μ(0),Σ(0)) and set the iteration counter *l* = 0 and set ε > 0.Step 2: For Φ(l)=(α(l),μ(l),Σ(l)), execute the following procedures:(E-step): Calculate    ψijl=αi(l)pi(zj;μi(l),Σi(l))p(zj;Φ(l)),j=1,2,⋯,N, i=1,2,⋯,s         Ψi(l)=∑j=1Nψij(l),i=1,2,⋯,s
(M-step): Update Φ(l)=α(l),μ(l),Σ(l) by computing
αi(l+1)=Ψi(l)N,i=1,2,⋯,sμi(l+1)=1Ψi(l)∑j=1Nψij(l)zj,i=1,2,⋯,s∑i(l+1)=1Ψi(l)∑j=1Nψij(l)(zj−μi(l+1))(zj−μi(l+1))T,i=1,2,⋯,sStep 3: If the prescribed convergence conditionmaxαi(l+1)−αi(l)αi(l),μi(l+1)−μi(l)μi(l),Σi(l+1)−Σi(l)Σi(l);≤ε, ε≪1  is satisfied, then set l∗=l+1 and exit. The optimal ML estimate of *Φ* is obtained by  Φ∗=Φ(l∗). Else set l=l+1 and go back to Step 2.

The above process is based on the premise of known submodels number. However, the submodels number *s* is unknown in advance. Thus, the estimation of *s* based on the information criteria related to the ML estimation can be further expressed as [39,40].

Firstly, two positive integers, *s*_min_ and *s*_max,_ are given to make the number of submodels in the interval [*s*_min_, *s*_max_]. 

Secondly, for all *s* = *s*_min_, …, *s*_max_, the parameter estimate *Φ_s_*, which denotes the estimate of *Φ* for a fixed *s*, is computed. Furthermore, the estimate of *s* can be expressed as follows
(16)$$\hat{s}=\underset{s=s_{\min},\cdots ,s_{\max}}{\arg\min}J(\Phi_{s},s)$$
where *J*(*Φ_s_*, *s*) denotes the criterion specified below. Based on the existed information criteria for model selection, the consistent Akaike’s information criterion (CAIC) [41] and the MDL criterion [42] are adopted in this work. The aforementioned criterion has the following form: (17)$$J(\Phi_{s},s)=-2L(\Phi_{s})+A(N)D(s)$$
(18)$$D(s)=(s-1)+s(n+p)+\frac{1}{2}s(n+p)(n+p+1)$$
(19)$$A(N)=\left\{\begin{array}{l} \ln N+1\;(CAIC)\\ \ln N\;\;\;(MDL) \end{array}\right.$$
*A*(*N*)*D*(*s*) denotes the penalization function of the data numbers and cluster numbers. *A*(*N*) denotes a function of the numbers of the data samples *N**. D*(*s*) denotes the number of independent parameters in *Φ_s_*. 

### 3.2. Affine Submodel Parameters Estimation

The parameters of each affine submodel need to be calculated in each cluster. The least squares method is adopted in this work as an accurate algorithm. *N* data samples are divided into *s* disjoint clusters. The number of data in the *i*th cluster is *N**_i_*. For the sample of *j**_iNi_*, the first subscript is the serial number of clusters, and the second subscript is the serial number of *N**_i_* samples in the *i*th cluster. Based on the descriptions, the equations and variables can be defined as follows [43,44]:(20)$$\sum_{i=1}^{s}N_{i}=N$$
(21)$$X_{i}={\left[{\bar{x}}_{j_{i1}}\;{\bar{x}}_{j_{i2}}\cdots {\bar{x}}_{j_{iN_{i}}}\right]}^{T}$$
(22)$${\bar{x}}_{j_{il}}=\left[\begin{array}{l} x_{j_{il}}\\ \;1 \end{array}\right]\;,l=1,2,\cdots ,N_{i}$$
(23)$$Y_{i}={\left[y_{j_{i1}}\;y_{j_{i2}}\;\cdots \;y_{j_{iN_{i}}}\right]}^{T}$$
(24)$$\Gamma_{i}=\left\{j_{i1},j_{i2},\cdots j_{iN_{i}}\right\}$$

On this basis, the parameters of each affine submodel can be estimated by the least square method:(25)$${\hat{\theta }}_{i}={\left(X_{i}^{T}X_{i}\right)}^{-1}X_{i}^{T}Y_{i}$$

### 3.3. Calculation of Hyperplane Coefficient Matrices

The hyperplane coefficient matrices need to be calculated to classify two adjacent clusters, *Γ_i_* and *Γ_j_*. The crucial step is to solve *s*(*s* − 1)/2 pattern recognition problems [45]. Thus, the improved proximal support vector machine (PSVM) method is adopted in this work [46].

Firstly, two adjacent clusters, *Γ_i_* and *Γ_j_*, can be obtained through the following equations:(26)$$\left\{\left.\Gamma_{i}\;,\;\Gamma_{j}\right\}=\right.\underset{2\leq i,j\leq s_{\max},i\neq j}{\min}\left\{\;\left.{\left\|\mu_{hi}-\mu_{hj}\right\|}^{2}\right\}\right.$$

Secondly, the parameters of the hyperplane surface are calculated by minimizing the objective function:(27)$$\begin{array}{l} \underset{(F,g,\xi )\in R^{n+1+p}}{\min}\frac{1}{2}\xi_{i}^{T}V_{i}\xi_{i}+\frac{1}{1+a_{i}}(F_{i}^{T}F_{i}+g_{i}^{2})\\ s.t.\;W_{i}(F_{i}x_{i}-eg_{i})=\lambda_{i}-\xi_{i} \end{array}$$
where *V_i_* and *W_i_* are diagonal matrices, *ξ_i_* is an error vector, and *e* is a unit vector. When *W_i_* = 1, *V_i_* = *v* (*v* is a penalty factor) and *λ_i_* = 1, while *W_i_* = −1, *V_i_* = *v*(*n*^+^/*n*^−^) and *λ_i_* = *a_i_* (*a_i_* is a positive number). *n^+^* denotes the number of positive samples, *n^−^* denotes the number of negative samples. 

To solve the problem in Equation (27), the following Lagrange equation can be constructed based on the *KKT* (Karush-Kuhn-Tucker) condition [47]:(28)$$L(F,g,\xi ,\eta )=\frac{1}{2}\xi_{i}^{T}V\xi_{i}+\frac{1}{1+a_{i}}(F_{i}^{T}F_{i}+g_{i}^{2})-\eta_{i}{}^{T}(W_{i}(F_{i}x_{i}-eg_{i})-\lambda_{i}+\xi_{i})$$
where *η_i_* is the Lagrange coefficient. The Lagrange’s conditional extremum is adopted in this work. Thus, the following equations can be further obtained:(29)$$\left\{\begin{array}{cc} \frac{\partial L(F,g,\xi ,\eta )}{\partial F}=0; & \frac{\partial L(F,g,\xi ,\eta )}{\partial g}=0\\ \frac{\partial L(F,g,\xi ,\eta )}{\partial \xi }=0; & \frac{\partial L(F,g,\xi ,\eta )}{\partial \eta }=0 \end{array}\right.$$

Then, it can be expressed as:(30)$$\left\{\begin{array}{l} F_{i}=\frac{a_{i}+1}{2}x_{i}^{T}W_{i}\eta_{i}\\ g_{i}=-\frac{1+a_{i}}{2}e_{i}{}^{T}W_{i}\eta_{i}\\ \xi_{i}=V_{i}{}^{-1}\eta_{i} \end{array}\right.$$

We can further get:(31)$$W_{i}(F_{i}x_{i}-g_{i}e_{i})=\frac{a_{i}+1}{2}W_{i}(x_{i}^{T}x_{i}+e_{i}{}^{T}e_{i})W_{i}\eta_{i}$$

Combining Equations (28) and (30), we can get:(32)$$\frac{a_{i}+1}{2}W_{i}(x_{i}^{T}x_{i}+e_{i}{}^{T}e_{i})W_{i}\eta_{i}=\lambda_{i}-V_{i}{}^{-1}\eta_{i}$$

Thus, we can get:(33)$$\eta_{i}={\left(V_{i}{}^{-1}+\frac{1+a_{i}}{2}\left(W_{i}\left(x_{i}x_{i}^{T}+e_{i}e_{i}^{T}\right)W_{i}\right)\right)}^{-1}\lambda_{i}$$

Based on Equation (33), the Lagrange coefficients can be obtained, and thus the hyperplane coefficient matrices, i.e., *F_i_* and *g_j_*, can then be calculated according to Equation (30).

Finally, the parameters of each sub-model are shown in Table 2 and based on this, the specific expression of the tire PWA model can be obtained as Equations (34) and (35). We can analyze that the tire longitudinal force is identified into 14 affine submodels, and the tire lateral force is identified into 10 affine submodels under combined conditions. Based on the different affine submodels in Table 2, the corresponding numbers are shown in the Figure 4.

Firstly, the simulation results are shown in Figure 4. Second, the fitting errors between the PWA model and the experimental data are shown in Figure 5. The distributions of fitting error are both concentrated near zero and the amplitudes of the fitting errors of the PWA model are relatively small compared with the experimental data. This illustrates that the identified PWA model can effectively reflect the nonlinear relationship. These results help to further verify the accuracy of the tire model in the PWA form.



(34)
$$\left\{\begin{array}{cc} y_{j}=\theta_{1}^{T}\left[\begin{array}{c} x_{j}\\ 1 \end{array}\right]=-177.8u_{k-1}^{1}+14936u_{k-1}^{2}-1713 & if\left[\begin{array}{ccc} 0.084 & -1 & -0.04\\ -0.176 & -1 & -0.112\\ 0.0046 & -1 & -0.139\\ -0.0087 & 1 & 0.0046\\ -1 & 0 & -12 \end{array}\right]{\left[x_{j}\;1\right]}^{T}\leq 0\\ \vdots & \vdots \\ y_{j}=\theta_{14}^{T}\left[\begin{array}{c} x_{j}\\ 1 \end{array}\right]=11.7u_{k-1}^{1}-607u_{k-1}^{2}-2746 & if\left[\begin{array}{ccc} -25.5 & -1 & 19.63\\ 0.061 & 1 & 0.626\\ 0.0083 & 1 & -0.525\\ 0 & -1 & -1\\ 1 & 0 & -12 \end{array}\right]{\left[x_{j}\;1\right]}^{T}\leq 0 \end{array}\right.$$





(35)
$$\left\{\begin{array}{cc} y_{j}=\theta_{1}^{T}\left[\begin{array}{c} x_{j}\\ 1 \end{array}\right]=-70.97u_{k-1}^{2}-4239u_{k-1}^{1}+1820 & if\;\;\;\left[\begin{array}{ccc} 0.249 & -1 & 1.06\\ -0.027 & -1 & 0.00055\\ -1 & 0 & -12\\ 0 & 1 & -0.5 \end{array}\right]{\left[x_{j}\;1\right]}^{T}\leq 0\\ \vdots & \vdots \\ y_{j}=\theta_{10}^{T}\left[\begin{array}{c} x_{j}\\ 1 \end{array}\right]=-81.9u_{k-1}^{2}+3309u_{k-1}^{1}-1680 & if\;\;\;\left[\begin{array}{ccc} -0.135 & -1 & 0.658\\ 0.0124 & -1 & -0.0433\\ 1 & 0 & -12\\ 0 & 1 & -0.5 \end{array}\right]{\left[x_{j}\;1\right]}^{T}\leq 0 \end{array}\right.$$



## 4. Vehicle State Estimation Based on Adaptive SCKF

Despite being an excellent estimation strategy, the Kalman filter algorithm also has some limitations. EKF linearizes the state function by calculating a Jacobian matrix. However, the truncation error is large when calculating a strong nonlinear system. The unscented transformation in the UKF is applied to approximate the probability density function more accurately than EKF. However, the accuracy and stability of UKF are reduced in the face of high-dimensional nonlinear systems. As a high-performance nonlinear filtering method, CKF has been widely studied in recent years [48]. A square-root cubature Kalman filter (SCKF) is proposed to ensure symmetry, positive (semi) qualitative analysis, and improve the estimation accuracy. When the measurement noise is unknown, the filtering accuracy and stability will be reduced [49,50]. For this reason, the MAP criterion is used to calculate the statistical values of process noise covariance and measurement noise covariance to improve the accuracy of state estimation. The overall framework for the estimation strategy is shown in Figure 6.

### 4.1. Nonlinear State Space Equation of Vehicle Model

In the existing studies, the longitudinal relaxation length is inevitably considered as a crucial factor in the estimation of longitudinal forces. The longitudinal relaxation length of the tire has a significant effect on the longitudinal dynamics modeling [51]. The dynamic equation of the tire’s longitudinal force can be written as follows:(36)$${\dot{F}}_{xij}=\frac{v_{x}}{\varepsilon_{xij}}(F_{xij\_M}-F_{xij})$$
where *ε_xij_* is the longitudinal slack length of each tire, *F_xij_M_* is the longitudinal force calculated by the PWA tire model, and *ε_xij_* represents the effect of the tire’s elastic hysteresis on the longitudinal force in the process of the tire–road interaction. In this work, the longitudinal relaxation length of the tire is constant and can be expressed as: *ε_x_* = *C_fx_/C_x_* where *C_fx_* is the longitudinal stiffness at the zero-point of the longitudinal force and *C_x_* is the longitudinal slip stiffness of the tire.

The PWA tire model established in this work can accurately reflect the tire’s nonlinear mechanical characteristics. The affine submodel switches between different sub-models according to different values of [sideslip angle *α*, longitudinal slip *λ*]. The expression of tire force is further simplified as follows:

(37)$$\left\{\begin{array}{cc} F_{xfl}=M_{i}^{x1}\alpha_{1}+N_{i}^{x1}\lambda_{1}+b_{i}^{x1}, & F_{xfr}=M_{i}^{x2}\alpha_{1}+N_{i}^{x2}\lambda_{2}+b_{i}^{x2}\\ F_{xrl}=M_{i}^{x3}\alpha_{3}+N_{i}^{x3}\lambda_{3}+b_{i}^{x3}, & F_{xrr}=M_{i}^{x4}\alpha_{3}+N_{i}^{x4}\lambda_{4}+b_{i}^{x4}\\ F_{yfl}=M_{j}^{y1}\alpha_{1}+N_{j}^{y1}\lambda_{1}+b_{j}^{y1}, & F_{yfr}=M_{j}^{y2}\alpha_{1}+N_{j}^{y2}\lambda_{2}+b_{j}^{y2}\\ F_{yrl}=M_{j}^{y3}\alpha_{3}+N_{j}^{y3}\lambda_{3}+b_{j}^{y3}, & F_{yrr}=M_{j}^{y4}\alpha_{3}+N_{j}^{y4}\lambda_{4}+b_{j}^{y4} \end{array}\right.$$
where the longitudinal force expression of each wheel is one of the longitudinal force affine submodel expressions, *i* = 1~14, and the lateral force expression of four wheels is one of the lateral force affine submodel expressions, *j* = 1~10. The sideslip angles of the two front wheels are considered approximately equal, defined as *α*_1_. Similarly, the sideslip angle of the rear wheels is defined as *α*_3_. *λ*_1_, *λ*_2_, *λ*_3_, and *λ*_4_ are the longitudinal slip on each tire respectively.

Through the analysis of vehicle dynamics, the sideslip angle of vehicle can be obtained as follows:(38)$$\dot{\beta }=\frac{1}{mv_{x}}\left(F_{yfl}+F_{yfr}+F_{yrl}+F_{yrr}\right)-r$$

The vehicle nonlinear state-space representation can be expressed as:(39)$$\left\{\begin{array}{l} \dot{x}(t)=f\left(x(t),u(t)\right)+w(t)\\ z(t)=h\left(x(t),u(t)\right)+v(t) \end{array}\right.$$

The state vector is defined as:(40)$$x={\left[r,\beta ,v_{x},v_{y},F_{xfl},F_{xfr},F_{xrl},F_{xrr}\right]}^{T}$$

The inputs vector is defined as:(41)$$u={\left[\delta ,\omega_{1},\omega_{2},\omega_{3},\omega_{4},a_{x},a_{y}\right]}^{T}$$

The measurement vector is shown as follows:(42)$$z={\left[r,a_{x},a_{y}\right]}^{T}$$

The equation of state and measurement equation can be expressed in the following form:(43)$$f(\cdot )=\left[f_{1},f_{2},f_{3},f_{4},f_{5},f_{6},f_{7},f_{8}\right]$$
(44)$$h(\cdot )=[h_{1},h_{2},h_{3}]$$

Therefore, the specific nonlinear function *f*(⋅) and *h*(⋅) can be derived as follows in Equations (45) and (46):

(45)$$\left\{\begin{array}{c} f_{1}=L_{1}x\left(2\right)+L_{2}\frac{x(1)}{x(3)}+L_{3}u\left(1\right)+L_{4}\frac{u(2)}{x(3)}+L_{5}\frac{u(3)}{x(3)}+L_{6}\frac{u(4)}{x(3)}+L_{7}\frac{u(4)}{x(3)}+L_{8}\\ f_{2}=L_{9}\frac{x(2)}{x(3)}+L_{10}\frac{x(1)}{x^{2}\left(3\right)}-x\left(1\right)+L_{11}\frac{u(1)}{x(3)}+L_{12}\frac{u(2)}{x^{2}\left(3\right)}+L_{13}\frac{u(3)}{x^{2}\left(3\right)}+L_{14}\frac{u(4)}{x^{2}\left(3\right)}+L_{15}\frac{u(5)}{x^{2}\left(3\right)}+L_{16}\frac{1}{x(3)}\\ f_{3}=u\left(6\right)+x\left(1\right)x\left(4\right);\\ f_{4}=u\left(7\right)-x\left(1\right)x\left(3\right);\\ f_{5}=L_{17}x\left(2\right)x\left(3\right)+L_{18}x\left(1\right)-L_{17}x\left(3\right)u\left(1\right)+L_{19}u\left(2\right)+L_{20}x\left(3\right)-L_{21}x\left(3\right)x\left(5\right)\\ f_{6}=L_{22}x\left(2\right)x\left(3\right)+L_{23}x\left(1\right)-L_{22}x\left(3\right)u\left(1\right)+L_{24}u\left(2\right)+L_{25}x\left(3\right)-L_{26}x\left(3\right)x\left(5\right)\\ f_{7}=L_{27}x\left(2\right)x\left(3\right)-L_{28}x\left(1\right)+L_{29}u\left(4\right)+L_{30}x\left(3\right)-L_{31}x\left(3\right)x\left(7\right)\\ f_{8}=L_{32}x\left(2\right)x\left(3\right)-L_{33}x\left(1\right)+L_{34}u\left(4\right)+L_{35}x\left(3\right)-L_{36}x\left(3\right)x\left(7\right) \end{array}\right.$$(46)$$\left\{\begin{array}{l} h_{1}=x\left(1\right)\\ h_{2}=\frac{1}{m}\left[\left(x(5)+x(6)\right)\cos u(1)-\left(F_{yfl}+F_{yfr}\right)\sin u(1)+x(7)+x(8)\right]\\ h_{3}=\frac{1}{m}\left[\left(x(5)+x(6)\right)\sin u(1)+\left(F_{yfl}+F_{yfr}\right)\cos u(1)+F_{yrl}+F_{yrr}\right] \end{array}\right.$$
where *L*_1_–*L*_36_ are the parameters derived from the dynamic analysis of the vehicle through the PWA tire model. The specific expressions are shown in Appendix A. According to the calculations for the sideslip angle and longitudinal slip of each wheel in preliminary processing, the affine submodels of longitudinal force and lateral force can be judged in real time.

### 4.2. The Standard SCKF Algorithm

To explain the proposed algorithm clearly, the standard SCKF algorithm is presented as a prerequisite. The discrete-time nonlinear system can be obtained as follow:(47)$$\left\{\begin{array}{l} x_{k}=f(x_{k-1},u_{k-1})+w_{k-1}\\ z_{k}=h(x_{k},u_{k})+v_{k} \end{array}\right.$$
where *x_k_* is the state vector of system and *z_k_* is the measurement vector of system. *f*(⋅) and *h*(⋅) are the nonlinear functions, which represent the state transition function and measurement function respectively. *w_k_*_−1_ and *v_k_* represent the system noise and measurement noise. The specific steps of standard SCKF algorithm can be summarized as follow [52]:

#### 4.2.1. Initialization

The state initial value ${\hat{x}}_{0|0}$ and the square-root factor *S*_0|0_ of the corresponding error covariance matrix can be set as follows:(48)$$S_{0|0}={\left[Chol(P_{0|0})\right]}^{T}$$
where *P*_0|0_ is the error covariance matrix and *Chol*(⋅) is the Cholesky decomposition.

#### 4.2.2. Time Update

The cubature points are calculated and transferred based on the state transition function:(49)$$\left\{\begin{array}{l} x_{k-1|k-1}^{i}={\hat{x}}_{k-1|k-1}+S_{k-1|k-1}\xi_{i},i=1,2,\cdots ,2n\\ x_{k|k-1}^{i*}=f\left(x_{k-1|k-1}^{i}\right) \end{array}\right.$$
where *S_k_*_−1|*k*−1_ is the square-root coefficient of *P_k_*_−1|*k*−1_ obtained by *Cholesky* decomposition, *S**_k_*_−1|*k*−1_ = [*Chol*(*P**_k_*_−1|*k*−1_)]^T^. *ξ_i_* is the column *i* of the cubature point weight matrix $\left[\sqrt{n}I_{n},-\sqrt{n}I_{n}\right]$, and *I_n_* is the unit matrix of *n* × *n*. *n* is the dimension of the state variable.


Calculate the predicted value of the state and the square-root factor of its error covariance matrix:

(50)
$$\left\{\begin{array}{l} {\hat{x}}_{k|k-1}=\frac{1}{2n}\sum_{i=1}^{2n}x_{k|k-1}^{i*}\\ S_{k|k-1}=Tria\left(\left[X_{k|k-1}^{*},Chol\left(Q_{k-1}\right)\right]\right) \end{array}\right.$$



*Tria*(⋅) stands for orthogonal triangular matrix factorization. The weighted central matrix $X_{k|k-1}^{*}$ is defined as:(51)$$X_{k|k-1}^{*}=\frac{1}{\sqrt{2n}}\left[x_{k|k-1}^{1*}-{\hat{x}}_{k|k-1},x_{k|k-1}^{2*}-{\hat{x}}_{k|k-1},\cdots ,\right.\left.x_{k|k-1}^{2n*}-{\hat{x}}_{k|k-1}\right]$$

#### 4.2.3. Measurement Update


The cubature points are updated using the state prediction value ${\hat{x}}_{k|k-1}$ and the square-root *S_k|k−_*_1_ of the prediction error covariance at *k* time. After that, it is transferred based on the measurement function as follow:

(52)
$$\left\{\begin{array}{l} x_{k|k-1}^{i}={\hat{x}}_{k|k-1}+S_{k|k-1}\xi_{i}\\ z_{k|k-1}^{i}=h\left(x_{k|k-1}^{i}\right) \end{array}\right.$$




The square-root factor of the measured predicted value and its innovation covariance matrix can be calculated as:

(53)
$$\left\{\begin{array}{l} {\hat{z}}_{k|k-1}=\frac{1}{2n}\sum_{i=1}^{2n}z_{k|k-1}^{i}\\ S_{k|k-1}^{zz}=Tria\left(\left[Z_{k|k-1},Chol\left(R_{k}\right)\right]\right) \end{array}\right.$$



The weighted central matrix *Z_k|k−_*_1_ is defined as:(54)$$Z_{k|k-1}=\frac{1}{\sqrt{2n}}\left[z_{k|k-1}^{1}-{\hat{z}}_{k|k-1},z_{k|k-1}^{2}-{\hat{z}}_{k|k-1},\cdots ,\right.\left.z_{k|k-1}^{2n}-{\hat{z}}_{k|k-1}\right]$$


The measurement covariance matrix and cross covariance matrix are calculated as follow:

(55)
$$\left\{\begin{array}{l} P_{k|k-1}^{zz}=S_{k|k-1}^{zz}{\left(S_{k|k-1}^{zz}\right)}^{T}\\ P_{k|k-1}^{xz}=X_{k|k-1}Z_{k|k-1}^{T} \end{array}\right.$$



The weighted central matrix *X_k|k−_*_1_ is defined as:(56)$$X_{k|k-1}=\frac{1}{\sqrt{2n}}\left[x_{k|k-1}^{1}-{\hat{x}}_{k|k-1},x_{k|k-1}^{2}-{\hat{x}}_{k|k-1},\cdots ,\right.\left.x_{k|k-1}^{2n}-{\hat{x}}_{k|k-1}\right]$$


Kalman gain matrix can be expressed as:

(57)
$$K_{k}=P_{k|k-1}^{xz}{\left(P_{k|k-1}^{zz}\right)}^{-1}$$




Update the square-root factor of the state variable and error covariance matrix at *k* time:

(58)
$$\left\{\begin{array}{l} {\hat{x}}_{k|k}={\hat{x}}_{k|k-1}+K_{k}\left(z_{k}-{\hat{z}}_{k|k-1}\right)\\ S_{k|k}=Tria\left(\left[x_{k|k-1}-K_{k}z_{k|k-1},K_{k}Chol\left(R_{k}\right)\right]\right) \end{array}\right.$$



### 4.3. Adaptive Square Cubature Kalman Filter (ASCKF)

It is assumed that process noise covariance *Q*, measurement noise covariance *R*, and system state *x_k_* are all unknown variables. Their MAP estimations can be obtained by maximizing the conditional density function:(59)$$J^{*}=p\left[X_{k},Q,R|Z_{k}\right]$$
where *X_k_* = {*x*_1_, *x*_2_, …, *x_k_*}, *Z_k_* = {*z*_1_, *z*_2_, …, *z_k_*}.

Based on the properties of conditional probability:$$p\left[X_{k},Q,R|Z_{k}\right]=\frac{p\left[X_{k},Q,R|Z_{k}\right]}{p\left[Z_{k}\right]}$$
then
(60)$$J^{*}=\frac{p\left[X_{k},Q,R|Z_{k}\right]}{p\left[Z_{k}\right]}$$
since *p*[*Z**_k_*] is independent of the maximization of *J**. Therefore, the problem of finding the extremum of the above equation can be transformed into finding the extremum of its molecule. The MAP estimates of *Q*, *R*, and *x_k_* can be obtained equivalently by the following maximization density functions:(61)$$\begin{array}{l} J=p\left[X_{k},Q,R,Z_{k}\right]\\ =p\left[Z_{k}|X_{k},Q,R\right]p\left[X_{k}|Q,R\right]p[Q,R] \end{array}$$
where *p*[*Q*,*R*] is obtained from prior information and can be regarded as a known constant. According to the probability multiplication rule:

(62)$$\begin{array}{cc} p\left[X_{k}|Q,R\right] & =p\left[x_{0}\right]\prod_{j=1}^{k}p\left[x_{j}|x_{j-1},Q\right]=\frac{\exp\left\{-\frac{1}{2}\left|\right|x_{0}-{\hat{x}}_{0}\left|\right|\;\;{\;}_{P_{0}^{-1}}^{2}\right\}}{{\left(2\pi \right)}^{\frac{n}{2}}{\left|P_{0}\right|}^{\frac{n}{2}}}\times \prod_{j=1}^{k}\frac{\exp\left\{-\frac{1}{2}\left|\right|x_{j}-f\left(x_{j-1}\right){\left|\right|}_{Q^{-1}}^{2}\right\}}{{\left(2\pi \right)}^{\frac{n}{2}}{\left|Q\right|}^{\frac{n}{2}}}\\ & =C_{1}{\left|P_{0}\right|}^{-\frac{1}{2}}{\left|Q\right|}^{-\frac{k}{2}}\exp\left\{-\frac{1}{2}\left[\left|\right|x_{0}-{\hat{x}}_{0}\left|\right|\;\;{\;}_{P_{0}^{-1}}^{2}+\sum_{j=1}^{k}\left|\right|x_{j}-f\left(x_{j-1}\right){\left|\right|}_{Q^{-1}}^{2}\right]\right\} \end{array}$$
where, *n* is the dimension of the state variable, and $C_{1}=1/{\left(2\pi \right)}^{n(k+1/2)}$ is a constant.

Assume that the measurements are known and independent of each other. Similarly, we can get:(63)$$\begin{array}{rr} p\left[Z_{k}|X_{k},Q,R\right] & =\prod_{j=1}^{k}p\left[z_{j}|x_{j},R\right]=\prod_{j=1}^{k}\frac{1}{{\left(2\pi \right)}^{\frac{m}{2}}|R{|}^{\frac{1}{2}}}\exp\left\{-\frac{1}{2}\left|\right|z_{j}-h\left(x_{j}\right){\left|\right|}_{R^{-1}}^{2}\right\}\\ & =C_{2}{\left|R\right|}^{-\frac{k}{2}}\exp\left\{-\frac{1}{2}\left|\right|z_{j}-h\left(x_{j}\right){\left|\right|}_{R^{-1}}^{2}\right\} \end{array}$$
where *m* is the dimension of the measured variable and $C_{2}=1/{\left(2\pi \right)}^{mk/2}$ is a constant. Substituting Equations (62) and (63) into Equation (61) this result is as follows:

(64)$$\begin{array}{cc} J & =C_{1}C_{2}{\left|P_{0}\right|}^{-\frac{1}{2}}{\left|Q\right|}^{-\frac{k}{2}}{\left|R\right|}^{-\frac{k}{2}}p\left[Q,R\right]\exp\left\{-\frac{1}{2}\left[\left|\right|x_{0}-{\hat{x}}_{0}\left|\right|{\;}_{P_{0}^{-1}}^{2}+\sum_{j=1}^{k}\left|\right|x_{j}-f\left(x_{j-1}\right){\left|\right|}_{Q^{-1}}^{2}+\right.\right.\left.\left.\sum_{j=1}^{k}\left|\right|z_{j}-h\left(x_{j}\right){\left|\right|}_{R^{-1}}^{2}\right]\right\}\\ \; & =C{\left|Q\right|}^{-\frac{k}{2}}{\left|R\right|}^{-\frac{k}{2}}\exp\left\{-\frac{1}{2}\left[\sum_{j=1}^{k}\left|\right|x_{j}-f\left(x_{j-1}\right){\left|\right|}_{Q^{-1}}^{2}+\left.\left.\sum_{j=1}^{k}\left|\right|z_{j}-h\left(x_{j}\right){\left|\right|}_{R^{-1}}^{2}\right]\right\}\right.\right. \end{array}$$
where
(65)$$C=C_{1}C_{2}{\left|P_{0}\right|}^{-\frac{1}{2}}p[Q,R]\times \exp\left\{-\frac{1}{2}\left|\right|x_{0}-{\hat{x}}_{0}{\left|\right|}_{P_{0}^{-1}}^{2}\right\}$$

The logarithm operation doesn’t change the extremum of the function, thus *J* and ln*J* have the same maximum value. Taking the logarithm of both sides of Equation (64), we can get

(66)$$\ln J=-\frac{k}{2}\ln|Q|-\frac{k}{2}\ln\left|R\right|-\frac{1}{2}\sum_{j=1}^{k}\left|\right|x_{j}-f\left(x_{j-1}\right){\left|\right|}_{Q^{-1}}^{2}-\frac{1}{2}\sum_{j=1}^{k}\left|\right|z_{j}-h\left(x_{j}\right){\left|\right|}_{R^{-1}}^{2}+\ln C$$
where ${\hat{x}}_{j-1|k}$ and ${\hat{x}}_{j|k}$ are assumed as known. The partial derivatives of the natural log of *J* with respect to *Q* and *R* is taken as follow:(67)$${\left.\frac{\partial \ln J}{\partial Q}\right|}_{Q={\hat{Q}}_{k-1}}^{x_{j-1}={\hat{x}}_{j-1|k},x_{j}={\hat{x}}_{j|k}}={\left.0;\;\frac{\partial \ln J}{\partial R}\right|}_{R={\hat{R}}_{k}}^{x_{j}={\hat{x}}_{j|k}}=0$$

It is difficult to achieve the estimation of ${\hat{x}}_{j-1|k}$ and ${\hat{x}}_{j|k}$. Thus, state estimation ${\hat{x}}_{j-1|j-1}$ and ${\hat{x}}_{j|j}$ or state prediction ${\hat{x}}_{j|j-1}$ can be used to replace them [53]. Furthermore, the suboptimal estimation of noise covariance *Q_k_*_−1_ and *R_k_* can be expressed as:(68)$${\hat{Q}}_{k-1}=\frac{1}{k}\sum_{j=1}^{k}\left\{\left[{\hat{x}}_{j|j}-f\left({\hat{x}}_{j|j-1}\right)\right]\times {\left[{\hat{x}}_{j|j}-f\left({\hat{x}}_{j|j-1}\right)\right]}^{T}\right\}$$



(69)
$${\hat{R}}_{k}=\frac{1}{k}\sum_{j=1}^{k}\left\{\left[z_{j}-h\left({\hat{x}}_{j|j-1}\right)\right]{\left[z_{j}-h\left({\hat{x}}_{j|j-1}\right)\right]}^{T}\right\}$$



The measurement innovation can be defined as:(70)$$\mu_{k}=z_{k}-{\hat{z}}_{k|k-1}$$

The equations can be obtained as follows:(71)$$\left\{\begin{array}{l} E\left[\mu_{k}\right]=E\left[z_{k}-{\hat{z}}_{k|k-1}\right]=0\\ E\left[\mu_{k}\mu_{k}^{T}\right]=E\left[\left(z_{k}-{\hat{z}}_{k|k-1}\right){\left(z_{k}-{\hat{z}}_{k|k-1}\right)}^{T}\right]=P_{k|k-1}^{zz} \end{array}\right.$$

Combining Equations (58) and (70):(72)$${\hat{x}}_{j|j}-{\hat{x}}_{j|j-1}=K_{j}\left(z_{j}-{\hat{z}}_{j|j-1}\right)=K_{j}\mu_{j}$$

$P_{j|j}=S_{j|j}S_{j|j}^{T}$ as a known condition. Compared the standard CKF algorithm with Equations (68) and (69), the mathematical expectation of noise covariance *Q_k_*_−1_ and *R_k_* is:



(73)
$$\begin{array}{cc} E\left[{\hat{Q}}_{k-1}\right] & =\frac{1}{k}\sum_{j=1}^{k}E\left\{\left[{\hat{x}}_{j|j}-f\left({\hat{x}}_{j|j-1}\right)\right]\times {\left[{\hat{x}}_{j|j}-f\left({\hat{x}}_{j|j-1}\right)\right]}^{T}\right\}=\frac{1}{k}\sum_{j=1}^{k}E\left[K_{j}\mu_{j}\mu_{j}^{T}K_{j}^{T}\right]\\ \; & \;=\frac{1}{k}\sum_{j=1}^{k}\left[P_{j|j-1}-P_{j|j}\right]=\frac{1}{k}\sum_{j=1}^{k}\left[P_{j|j-1}^{xx^{*}}-P_{j|j}+Q_{j-1}\right]\neq Q_{k-1} \end{array}$$


(74)
$$E\left[{\hat{R}}_{k}\right]=\frac{1}{k}\sum_{j=1}^{k}E\left\{\left[z_{j}-h\left({\hat{x}}_{j|j-1}\right)\right]{\left[z_{j}-h\left({\hat{x}}_{j|j-1}\right)\right]}^{T}\right\}=\frac{1}{k}\sum_{j=1}^{k}P_{j|j-1}^{zz*}+R_{j}\neq R_{k}$$



(75)$$P_{j|j-1}^{xx^{*}}=\frac{1}{2n}\sum_{i=1}^{2n}\left[x_{j|j-1}^{i}-{\hat{x}}_{j|j-1}\right]{\left[x_{j|j-1}^{i}-{\hat{x}}_{j|j-1}\right]}^{T}$$(76)$$P_{j|j-1}^{zz^{*}}=\frac{1}{2n}\sum_{i=1}^{2n}\left[z_{j|j-1}^{i}-{\hat{z}}_{j|j-1}\right]{\left[z_{j|j-1}^{i}-{\hat{z}}_{j|j-1}\right]}^{T}$$
when the noise statistics change is small [54], the following equation can be considered valid: *Q_k_*_−1_ = *Q_j_*_−1_ and *R_k_* = *R_j_*. Rewrite Equations (73) and (74), we can get:



(77)
$$E\left[{\hat{Q}}_{k-1}\right]=\frac{1}{k}\sum_{j=1}^{k}E\left[K_{j}\mu_{j}\mu_{j}^{T}K_{j}^{T}\right]=\frac{1}{k}\sum_{j=1}^{k}\left[P_{j|j-1}^{xx^{*}}-P_{j|j}+Q_{k-1}\right]$$


(78)
$$E\left[{\hat{R}}_{k}\right]=\frac{1}{k}\sum_{j=1}^{k}E\left[\mu_{j}\mu_{j}^{T}\right]=\frac{1}{k}\sum_{j=1}^{k}P_{j|j-1}^{zz*}+R_{k}$$



Therefore, the statistical estimation of noise covariance *Q_k_*_−1_ and *R_k_* can be obtained as:(79)$${\hat{Q}}_{k-1}=\frac{1}{k}\sum_{j=1}^{k}\left[K_{j}\mu_{j}\mu_{j}^{T}K_{j}^{T}-P_{j|j-1}^{xx^{*}}+P_{j|j}\right]$$
(80)$${\hat{R}}_{k}=\frac{1}{k}\sum_{j=1}^{k}\left[\mu_{j}\mu_{j}^{T}-P_{j|j-1}^{zz*}\right]$$

## 5. The Simulation Analysis

In this section, the precision and effectiveness of the proposed estimation strategy under critical maneuvers were verified using a co-simulation platform based on high-fidelity software CarSim and Simulink. Two different simulation conditions, Case 1 and Case 2, were designed. The sine steering angle input of the autonomous vehicles was employed in Case 1. The vehicle speed and road adhesion coefficient were 120 km/h and 0.8. In Case 2, the combined condition was set at a medium speed and large steering angle. In addition, another estimation strategy designed based on the SCKF algorithm was selected for comparison. The vehicle parameters adopted in the simulation are given in Table 3.

Case 1: Simulation Results of Sine Steering Angle Input

The sine steering angle input is shown in Figure 7. Figure 8 shows the estimation results of the autonomous vehicle under different estimation strategies compared with the CarSim simulation results. Figure 8a,b show that the proposed estimation algorithm has better accuracy than the SCKF algorithm. The simulation results in CarSim are considered the real motion state of the autonomous vehicle. The general trend of the SCKF algorithm is like the CarSim simulation results, but its accuracy is lower than the ASCKF algorithm. Figure 8c shows the longitudinal vehicle velocity of the two estimation algorithms. It is obvious that the proposed strategy has better stability in the co-simulation. The peak error of the adaptive SCKF algorithm is relatively smaller and tends to stabilize quickly. This indicates that the adaptive SCKF has the ability to self-adapt when dealing with uncertainty interference.

Figure 8e–h show the simulation results of four-wheel longitudinal force. The tire model in this work is established by experimental data and has the advantage of concise form. Combined with the adaptive algorithm proposed in this paper, the simulation result has better precision and better immunity to unknown noise. In this case, the tire’s mechanical characteristics are easy to reach in the nonlinear region, and the vehicle system is highly nonlinear. It is significant to estimate the four-wheel longitudinal force stably and accurately for vehicle trajectory tracking and lateral stability control.

Case 2: Simulation Results of *J* turn Input

In Case 2, the *J* turn maneuver is set as the varying vehicle velocity and the given steering angle shown in Figure 9. In the typical scenario, the speed of the autonomous vehicle is increased from 30 km/h to 60 km/h and the road adhesion coefficient is set to *μ* = 0.6.

Figure 10 shows the simulation comparison results of Case 2. As can be seen from Figure 10a,b, the yaw rate and sideslip angle estimation results of the ASCKF estimation strategy proposed in this work has higher precision and anti-interference ability than the SCKF strategy. In the whole simulation, the proposed SCKF algorithm can adaptively adjust the measurement noise covariance to ensure the tracking accuracy of state estimation. In the estimation results of longitudinal velocity and lateral velocity in Figure 10c,d, the simulation results of the ASCKF algorithm converges faster to the CarSim simulation results.

According to Figure 10e–h, the error of the longitudinal force estimation is larger without the adaptive SCKF algorithm when the vehicle accelerates. When the estimation of the statistical characteristics of noise is not accurate enough, it will lead to the mutation of error. The adaptive algorithm based on MAP estimation can effectively improve this condition and converges to be stable quickly. It indicates that the adaptive SCKF algorithm has a better performance of dynamic adjustment when dealing with uncertainty interference. The high-precision and stable estimation of the four-wheel longitudinal force is the premise of autonomous vehicle dynamics control. It proves the anti-interference ability and effectiveness of the ASCKF algorithm in this work.

## 6. Conclusions

This paper proposes an adaptive estimation strategy based on a novel tire model in the PWA form that considers tire nonlinear mechanical characteristics for the autonomous vehicle to improve estimation accuracy and stability under critical maneuvers. Firstly, the PWA identification method, which mainly involves the data clustering, the parameter estimation of the affine submodels, and the calculation of the hyperplane coefficient matrices, is used to realize the modeling of the tire nonlinear mechanical characteristics under combined conditions. Secondly, a novel nonlinear state function that considers the tire longitudinal relaxation length and includes the PWA tire model is proposed. Finally, an adaptive square-root cubature Kalman filter estimation strategy based on the MAP criterion is applied in this work to estimate yaw rate, sideslip angle, longitudinal vehicle speed, lateral vehicle speed, and four-wheel longitudinal force. The CarSim-Simulink co-simulation results show that the ASCKF algorithm proposed in this work still maintains higher accuracy and stability against the other estimate strategy when the state changes suddenly or the statistical characteristics of noise are unknown. It shows the effectiveness and practicability of the novel estimation architecture proposed in this work. Future works will focus on the experimental assessment of the proposed estimation strategy on a real-world FWIA vehicle.

## Figures and Tables

**Figure 1 sensors-22-03403-f001:**
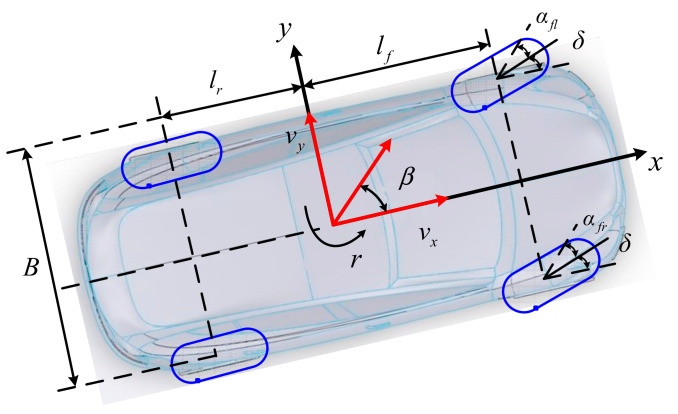
The vehicle model: top view.

**Figure 2 sensors-22-03403-f002:**
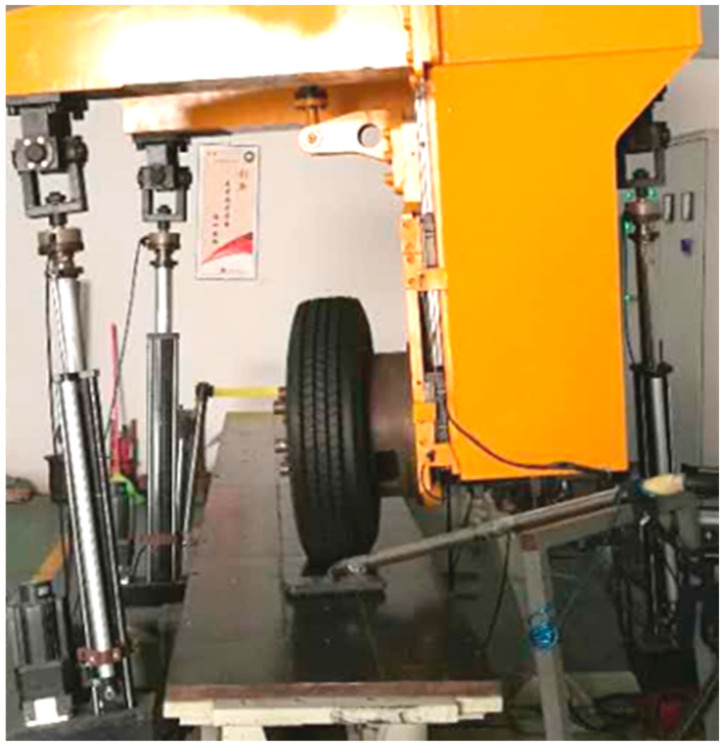
Bench test scenario of the tire mechanical properties.

**Figure 3 sensors-22-03403-f003:**
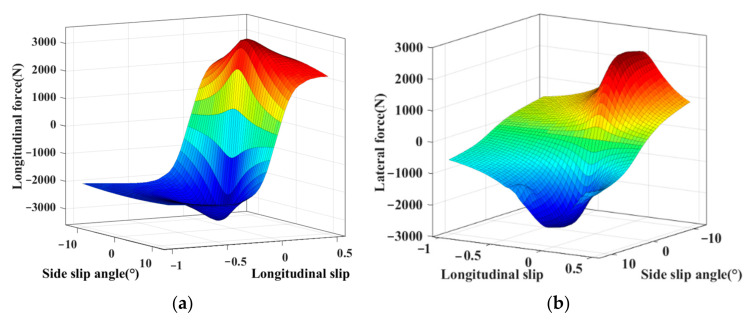
(**a**) Nonlinear relationship between the tire longitudinal force and its influence factors; (**b**) Nonlinear relationship between the tire lateral force and its influence factors.

**Figure 4 sensors-22-03403-f004:**
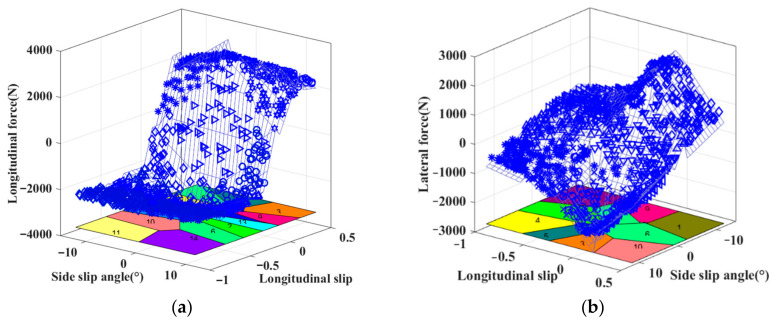
(**a**) Simulation result of the identified PWA model for approximating the nonlinear relationship between the tire longitudinal force and its influence factors; (**b**) Simulation result of the identified PWA model for approximating the nonlinear relationship between the tire lateral force and its influence factors.

**Figure 5 sensors-22-03403-f005:**
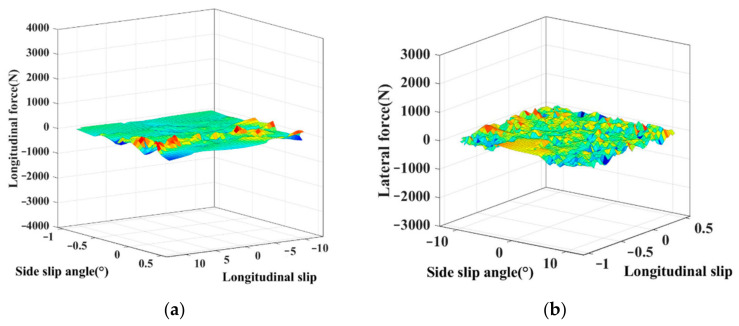
(**a**) Tire longitudinal force error between the PWA model and the experimental data; (**b**) Tire lateral force error between the PWA model and the experimental data.

**Figure 6 sensors-22-03403-f006:**
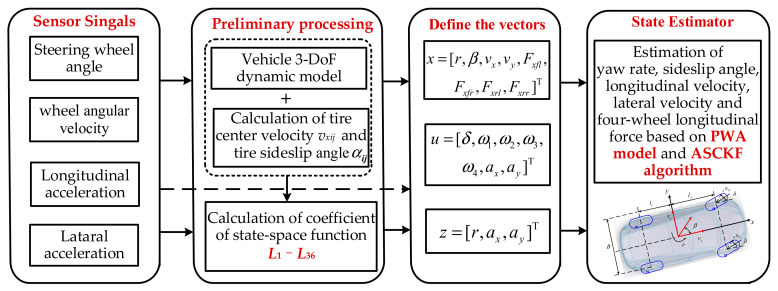
The overall framework for the estimation strategy.

**Figure 7 sensors-22-03403-f007:**
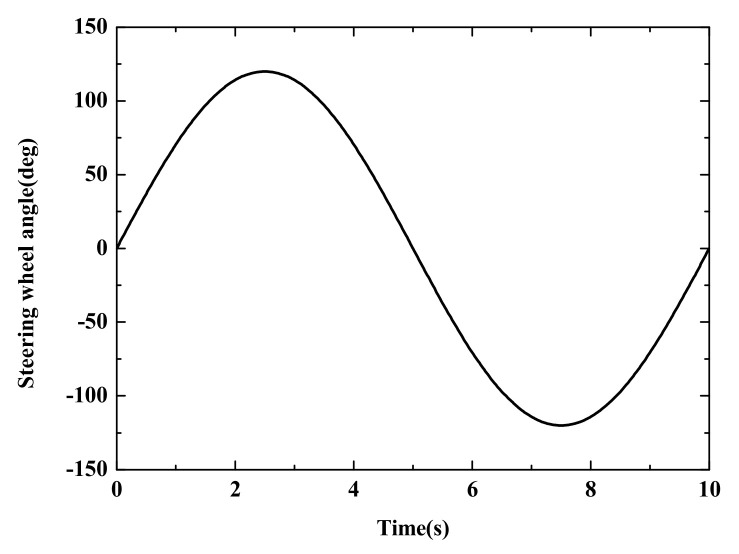
The Steering angle.

**Figure 8 sensors-22-03403-f008:**
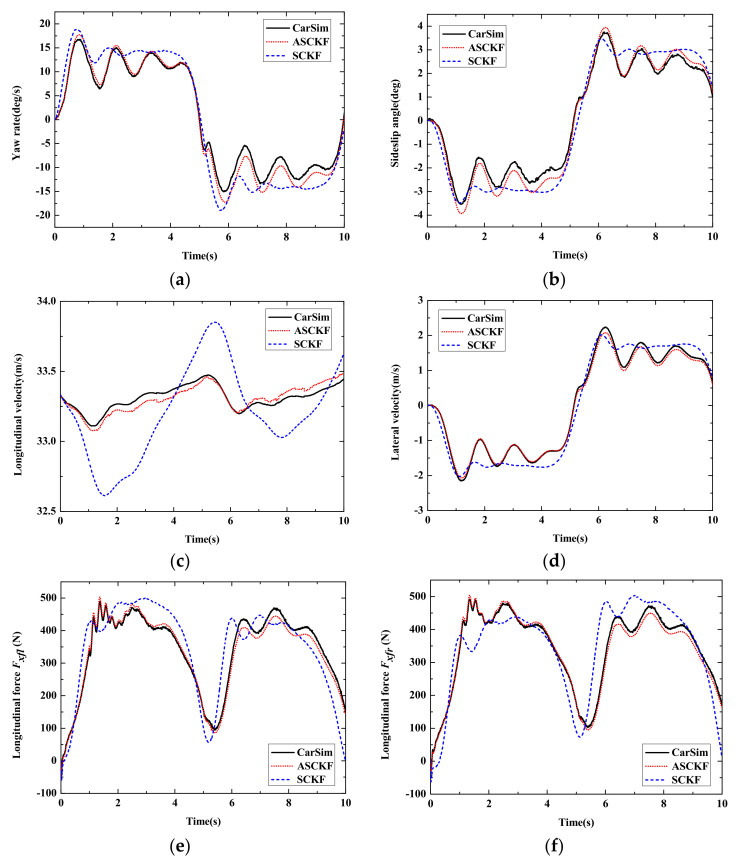
Estimation results of Case1. (**a**)Yaw rate; (**b**) Sideslip angle; (**c**) Longitudinal vehicle speed; (**d**) Lateral vehicle speed; (**e**) Longitudinal force *F_fl_*; (**f**) Longitudinal force *F_fr_*; (**g**) Longitudinal force *F_rl_*; (**h**) Longitudinal force *F_rr_*.

**Figure 9 sensors-22-03403-f009:**
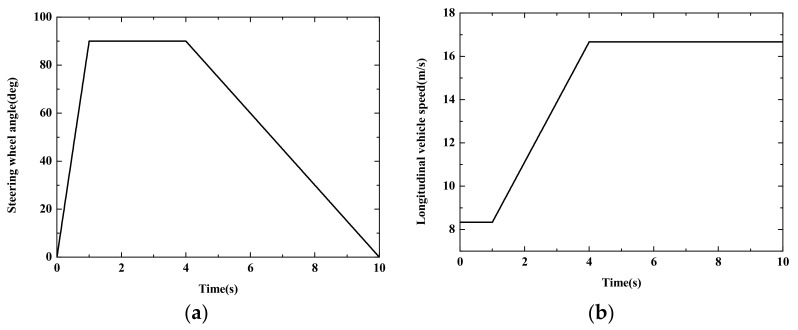
Simulation results of Case 2. (**a**) Steering wheel angle; (**b**) Longitudinal vehicle speed.

**Figure 10 sensors-22-03403-f010:**
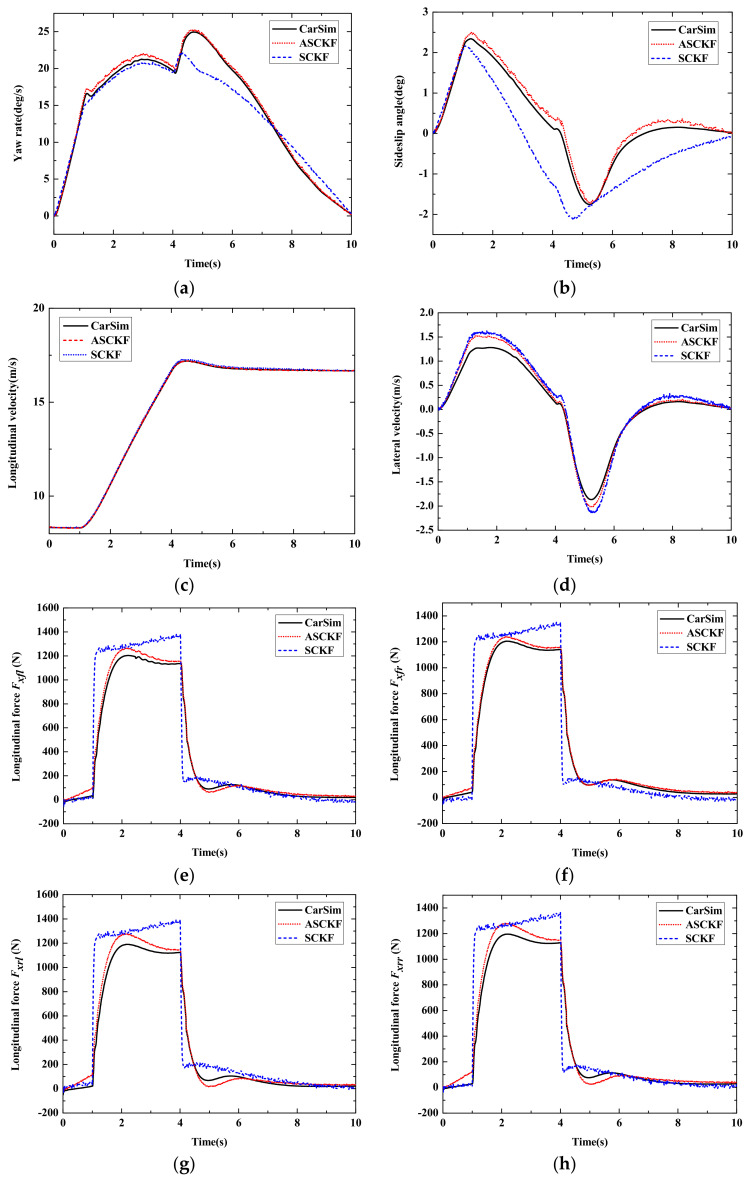
Simulation results of Case 2. (**a**) Yaw rate; (**b**) Sideslip angle; (**c**) Longitudinal vehicle speed; (**d**) Lateral vehicle speed; (**e**) Longitudinal force *F_fl_*; (**f**) Longitudinal force *F_fr_*; (**g**) Longitudinal force *F_rl_*; (**h**) Longitudinal force *F_rr_*.

**Table 1 sensors-22-03403-t001:** Bench test scenario of the tire mechanical properties.

Parameter	Setting
Tire pressure (kPa)	880
Tire vertical load (N)	8060
Tire sideslip angle (°)	−10~10
Tire longitudinal slip	−1~0.5
Road adhesion coefficient	0.34
Tire camber angle (°)	0
Velocity of rolling plate (mm/s)	200

**Table 2 sensors-22-03403-t002:** Parameters of each affine submodel.

PWA Model	Affine Submodel Parameters		
Tire longitudinal force	−177.8	14,936	−1713
	112.2	−163	−3266
	−82.1	−804	3283
	175.1	13,775	1784
	−0.58	−2513	3614
	30.3	−1533	−3332
	−12.2	37,666	46.5
	120.8	170	3292
	−191	9411	2267
	−34	−986	−3088
	−11.9	−536	−2686
	−119.1	−111	−3346
	−193.7	16,750	−1590
	11.7	−607	−2746
Tire lateral force	−70.97	−4239	1820
	−87.82	154.6	124
	−14.83	−3887	−2338
	−66.3	−630	−773
	−86.8	−2316	−1285
	−340	−141.4	8.5
	−56	586	476
	−165.3	2686	981
	−48.9	4741	2012
	−81.9	3309	−1680

**Table 3 sensors-22-03403-t003:** Simulation parameters.

Parameter	Setting
*m* (kg)	2350
*R* (m)	0.25
*l_f_* (m)	1.337
*l_r_* (m)	1.587
*B* (m)	1.53
*I_z_* (kg·m^2^)	4386
*h_g_* (m)	0.652

## Data Availability

Not applicable.

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
