# Peer review of "Estimation of Longitudinal Force, Sideslip Angle and Yaw Rate for Four-Wheel Independent Actuated Autonomous Vehicles Based on PWA Tire Model"

_sensors, 2022, doi:10.3390/s22093403_

Round 1

Reviewer 1 Report

Dear Authors,

The paper presents in-depth approach to estimation of dynamic characteristics of a tyre basing on a new tyre model. I have a few questions and comments regarding the study that, in my opinion, shoud be adressed to:

  • What is the vehicle model used for comparison in paragraph '4. The simulation analysis'? Is it the same model which is depicted in paragraph 2.? What are its paremeters' values (dimensions, inertia, mass, etc.)?
  • More detailed information should be provided on the test tire's parameters, at least its size and load index. 
  • Where there any torques applied to the vehicle model's wheels, and thus any control strategy during simulation? Or simulation was carried out without applying torques, providing only initial velocities?
  • Chcek for spelling (e.g. Tucker instead for Tucher) and other language issues, such as: When the suddenly change, etc. 

Author Response

We have modified our paper in detail according to your comments. The revisions of our manuscript are in accordance with submission requirements.

Reviewer 2 Report

The manuscript studies an interesting work to derive an estimation framework for four-wheel independently actuated autonomous vehicles. The two major contributions of this work are in developing the estimation framework and in developing the new tyre model. First, the methodology followed to derive the tyre model is good and appropriate for this study. The results also show a good agreement of this model with the experiments. Secondly, the Kalman filter-based estimation strategy (ASKCF) to estimate yaw rate, sideslip angle, lateral and longitudinal vehicle speed is shown to match well. I would recommend the paper to be published after addressing the following comment.

  • Why is the new tyre model important? It is proven from the results that the new tyre model matches the experimental results, thus correct. However, is it better than existing models? The authors should comment on this. 

Author Response

(The authors gave the same response as above.)

Reviewer 3 Report

  1. Improve the presentation quality of the EM algorithm (Figure 4).

  2. Standardize the number of decimals in Table 2.

  3. Make correction in: “The overall framework for estimation strategy is shown in Figure 9” to “... Figure 7”.

  4. Check punctuation at the end of sentences. For example in: “... can be used to replace them. [53]” .

  5. Improve the quality of Figures 8 to 11.

  6. Restructure the Conclusion section. Indicate the main results and possibilities for future work.

Author Response

(The authors gave the same response as above.)

Reviewer 4 Report

Once again proposed paper more related to an engineering tool report than a scientific paper. There is no comparison with state of the art approaches in metamodelisation, as well as IRL expected computational power requested.

The structure of the document is hard to read, for exemple Figure 4 is impossible to catch. Many (the majority) of 'bulk' equations have to be described in an annexe so that not to confuse the reader, for exemple equations (46). Go straight to the point instead in the main core article !

I think the proposed article has to be entirely rewritten

Author Response

(The authors gave the same response as above.)
